# Estimating the Standardized Precipitation Evapotranspiration Index Using Data-Driven Techniques: A Regional Study of Bangladesh

Ahmed Elbeltagi [1,†], Faisal AlThobiani [2], Mohammad Kamruzzaman [3,*,†], Shamsuddin Shaid [4], Dilip Kumar Roy [5], Limon Deb [6], Md Mazadul Islam [7], Palash Kumar Kundu [8] and Md. Mizanur Rahman [3]

1   Agricultural Engineering Department, Faculty of Agriculture, Mansoura University, Mansoura 35516, Egypt; ahmedelbeltagy81@mans.edu.eg
2   Faculty of Maritime Studies, King Abdulaziz University, Jeddah 21589, Saudi Arabia; falthobiani@kau.edu.sa
3   Farm Machinery and Postharvest Technology Division, Bangladesh Rice Research Institute, Gazipur 1701, Bangladesh; mizan.brri2015@gmail.com
4   School of Civil Engineering, Faculty of Engineering, Universiti Teknologi Malaysia (UTM), Johor Bahru 81310, Malaysia; sshahid@utm.my
5   Irrigation and Water Management Division, Bangladesh Agricultural Research Institute, Gazipur 1701, Bangladesh; dilip.roy@my.jcu.edu.au
6   Agricultural Economics Division, Bangladesh Rice Research Institute, Gazipur 1701, Bangladesh; limondeb.bau@gmail.com
7   Tuber Crops Research Centre, Bangladesh Agricultural Research Institute, Gazipur 1701, Bangladesh; mazadul.islam@bari.gov.bd
8   Irrigation and Water Management Division, Bangladesh Rice Research Institute, Gazipur 1701, Bangladesh; kundu22_bau@yahoo.com
*   Correspondence: milonbrri@gmail.com; Tel.: +880-177-6422-808
†   These authors contributed equally to this work.

**Abstract:** Drought prediction is the most effective way to mitigate drought impacts. The current study examined the ability of three renowned machine learning models, namely additive regression (AR), random subspace (RSS), and M5P tree, and their hybridized versions (AR-RSS, AR-M5P, RSS-M5P, and AR-RSS-M5P) in predicting the standardized precipitation evapotranspiration index (SPEI) in multiple time scales. The SPEIs were calculated using monthly rainfall and temperature data over 39 years (1980–2018). The best subset regression model and sensitivity analysis were used to determine the most appropriate input variables from a series of input combinations involving up to eight SPEI lags. The models were built at Rajshahi station and validated at four other sites (Mymensingh, Rangpur, Bogra, and Khulna) in drought-prone northern Bangladesh. The findings indicated that the proposed models can accurately forecast droughts at the Rajshahi station. The M5P model predicted the SPEIs better than the other models, with the lowest mean absolute error (27.89–62.92%), relative absolute error (0.39–0.67), mean absolute error (0.208–0.49), root mean square error (0.39–0.67) and highest correlation coefficient (0.75–0.98). Moreover, the M5P model could accurately forecast droughts with different time scales at validation locations. The prediction accuracy was better for droughts with longer periods.

**Keywords:** drought prediction; standardized precipitation evapotranspiration index; hybrid machine learning; additive regression; northern Bangladesh

## 1. Introduction

Drought is one of the most complicated recurring natural disasters, defined by a deficiency of precipitation, causing prolonged water scarcity. Failure to manage drought risk effectively has the potential to have dire consequences for people, livelihoods, the economy, and ecosystems [1–6]. Like the rest of South Asia, Bangladesh is plagued by periodic droughts. Due to the adverse impact on agricultural productivity and the environment,

it is one of Bangladesh's most expensive natural disasters [7,8]. Drought-related damage to agriculture is more prevalent than any other natural disaster in the country. Pre- and post-monsoon droughts (March–May and October–November) are the most common in Bangladesh, and pre-monsoon droughts can last into the monsoon season, delaying the arrival of rain. Drought susceptibility, on the other hand, varies according to region. In recent years, annual rainfall has increased across Bangladesh, including in the drought-prone northern region, while the western region has seen a decrease [2,9]. Extreme climate events such as droughts have increased in frequency over the past two decades, which has put food security in some regions at risk [10,11]. The frequency of extreme droughts will nearly double in the coming decades [12]. Thus, drought forecasting, and early warning are critical for agricultural resilience to climate change.

Due to its complexity and spatial–temporal development, drought forecasting is one of the most difficult issues for climate scientists and hydrologists [13]. Statistical, dynamical [14,15], and hybrid models [16] are typically utilized to forecast drought occurrences. For drought forecasting, statistical models employ correlation relationships between climate variables and drought indicators [17]. As opposed to statistical models, dynamical models are founded on the physical connections between the earth, ocean, and climate. These interactions are theoretically defined and resolved in dynamic models to develop drought forecasts and predictions [18]. A hybrid model, in contrast, is a combination of statistical and dynamical techniques [19,20]. For instance, to create an ensemble prediction, multiple dynamical model predictions can be combined using a statistical framework that assigns weights to the various dynamical model predictions [21]. Due to their simplicity [22] and low processing costs, statistical models are widely used to predict droughts [23–25]. Recently, machine learning (ML) techniques have also been used to predict drought in many global regions.

ML strategies involve a set of commands that enable systems to learn and improve without extensive programming [26–29]. In different climatological applications, such as rainfall prediction, ML algorithms have been used to develop models that can reproduce the empirical relationships between the variables [30], drought prediction [31], forecasting heat waves [32], and runoff simulation [17]. There are many critical issues to consider when developing a drought forecasting model. Systematic monitoring and early warning of impending droughts are essential for effective drought management. Various statistical models are widely used to predict droughts, such as ARIMA, multilinear regression (MLR), and the Markov chain [33]. As a result, hydrological research typically employs non-linear time series models. When it comes to predicting drought and climate, ML applications have performed exceptionally well in recent years [34,35]. SVR, RF, ELM, M5T, ANFIS, LGP, ERT, LSSVR, and MVR have all been successfully applied to drought forecasting [22,36–49]. Several recent studies have shown that hybrid ML models perform exceptionally well and accurately [22,42,50–53]. However, it is not always the case. Thus, evaluation of the models is essential to predict drought in a particular region.

Additionally, models for drought forecasting exist in different parts of the world, but a universal or ideal model cannot be developed for all climates. Incorrect model structure variables can lead to erroneous predictions. Thus, it is necessary to evaluate the models at the regional scale. Only a few studies have been performed using ML algorithms to predict drought in Bangladesh. SPI has been used in previous studies to predict droughts [44,54]. Despite this, no research has been conducted in Bangladesh on the use of ML and hybrid methods to predict SPEI. This study's primary goal was to fill in this knowledge gap.

Compared to other parts of the country, the Barind tract and the Teesta floodplain regions of the northern and northwestern parts (known as North Bengal) are highly impacted by drought due to high poverty rates, dependency on agriculture, low adaptive capacity and high variability of annual and seasonal rainfall [55,56]. Drought is a recurrent event in these regions [1,57]. Over the years, drought severity, frequency, and variability have increased in North Bengal [1,2,7,8,58–60]. Several studies indicated that droughts have significantly affected agricultural production and the natural environment in the

country in recent years [7,61]. Although there have been tremendous improvements in irrigation systems in Bangladesh in recent decades, agricultural activities remain dependent on seasonal rainfall [62]. A study conducted by Islam [63] indicated that North Bengal has experienced significant increases in rainfall variability, long seasonal-scale dry spells and numerous instances of below-normal rainfall in recent decades, significantly hampering crop growth. In addition, variability in temperature has a substantial effect on crop yields (such as rice and wheat) in North Bengal [64]. However, there is a dearth of literature regarding the evolution of drought in eastern Bangladesh.

Drought analysis and prediction need reliable rainfall and temperature data recorded for a longer period. However, an uninterrupted climate record for a longer period is not available in most of the meteorological stations of Bangladesh. This limits drought prediction using conventional statistical and numerical models. ML algorithms could be one of the solutions for bridging the climate data gap. Furthermore, to the best of our knowledge, no research has been published in the literature that predicts the SPEI in the study region using ML approaches applied. Hence, this work aimed to assess the performance of standalone models and novel hybrid ML algorithms in SPEI estimation. Additive regression, random subspace, and M5P tree models and their hybridized versions were used to predict SPEI. Agricultural drought is measured in Bangladesh using a 6-month SPEI, while the water scarcities, river flow declines, and hydrological droughts are measured using 9, 12, and 24-month SPEIs. As a result, forecasting models were developed to predict SPEI for those four timescales over 38 years (1980–2018). The models were developed for a single station in northwestern Bangladesh (Rajshahi), which is extremely prone to droughts, and tested at four locations, including Bogra, Rangpur, Mymensingh and Khulna.

The organization of the remaining parts of this article is as follows. Section 2 explains the study area and the data used in the study. Additionally, the theories of the ML algorithms used and the construction of prediction models using ML are discussed in this section. Section 3 explains the results obtained in this study. Section 4 provides a discussion of the study's findings. Finally, conclusions drawn from the results and discussion are provided in Section 5.

## 2. Materials and Methods

### 2.1. Study Area

Bangladesh covers a land area of 147,570 km$^2$ and is situated between latitudes 20.34° N and 26.38° N and longitudes 88.01° E to 92.41° E. It is surrounded by the deltas of several rivers that originate in the Himalayas (Figure 1). A humid tropical climate characterizes most of the country. The temperature is below 12.8 °C in January and above 31.1 °C in May. The temperature's spatial variability is extremely low because of the country's extremely flat topography. Different seasons have different temperature gradient orientations. As a result, the yearly average temperature in different regions is very similar. Rainfall in Bangladesh varies between northwest and northeastern Bangladesh by 1600 and 4400 mm, respectively. Seasonal and yearly variations in rainfall are significant. Nearly 70% of the year's rainfall occurs between May and September, with only 3% occurring in the winter months between December and February. In large parts of the country, the annual monsoon rainfall variability coefficient is greater than 30%. Since the country experiences a wide range of rainfall variability, droughts are common. Following its independence in 1973, Bangladesh was hit by droughts in 1978, 1979, 1981, 1982, 1989, 1992, 1994, and 1995. In recent memory, the worst droughts were in 1973, 1979, and 1994–1995, when the northwestern region lost 3.5 million tons of rice alone. A 25–30% decrease in average crop production was recorded in this region during the drought of 2006 [65,66].

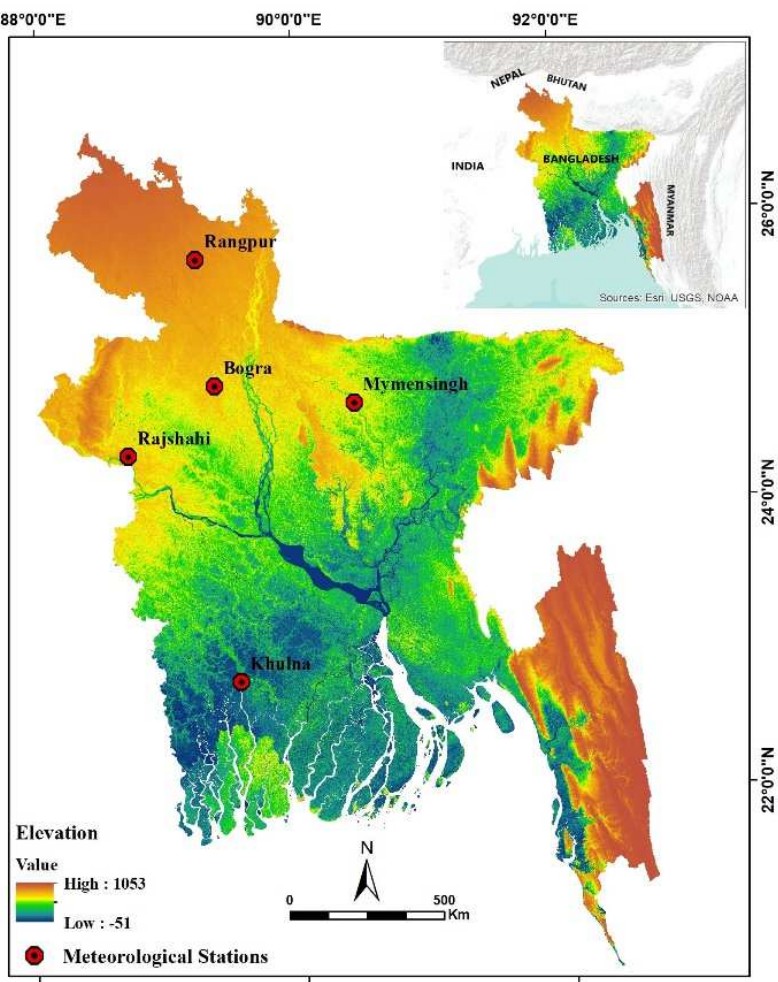

**Figure 1.** Study area and weather stations.

*2.2. Data*

Monthly temperature and precipitation data were collected from five meteorological stations (Table 1) from the Bangladesh Agricultural Research Council (BARC) for a 38-year period (1980–2018). The BARC team checked all the datasets for errors and inconsistencies. The rainfall and temperature data were seasonally divided using the R function na.seasplit [67] to examine missing data in the monthly climate variables. No missing data were found at these sites during the research period.

**Table 1.** Meteorological stations and descriptive statistics for rainfall and temperature.

| Station Name | Geographical Locations (Lat × Lon) | Elevation MSL (m) | Annual Mean Rainfall (mm) | Tmax (°C) | Tmin (°C) | Remarks |
|---|---|---|---|---|---|---|
| Rajshahi | 24.37 × 88.7 | 19.5 | 1424.51 | 31.31 | 20.50 | Model Development station |
| Bogra | 24.85 × 89.37 | 17.9 | 1738.05 | 30.75 | 21.01 | Model Validation station |
| Mymensingh | 90.43 × 24.72 | 18 | 2264.20 | 29.90 | 20.88 | Model Validation station |
| Khulna | 89.53 × 22.78 | 3.6 | 1834.92 | 31.28 | 21.77 | Model Validation station |
| Rangpur | 89.23 × 25.73 | 32.61 | 2248.82 | 29.63 | 20.24 | Model Validation station |

*2.3. Methodology*

Data were collected in the first phase to estimate the SPEI from 1980 to 2018. The best subset regression model for selecting the optimal combination of climatic variables was then used to create seven machine learning models (standalone and hybrid) to predict

the multiscale SPEI. The validation stations were used to forecast SPEIs using the best prediction model.

### 2.3.1. SPEI Calculation

Standardized precipitation evapotranspiration indexes (SPEIs) were derived from precipitation and temperature data as part of a simple water balance to account for surface evaporation changes, which is more sensitive to drought reactions caused by rising global temperatures.

The difference in the water balance was normalized as a log-logistic probability distribution in order to estimate SPEI's value. In mathematical terms, the probability density function is represented by the following equation:

$$f(x) = \frac{\beta}{\alpha}\left(\frac{x-\lambda}{\alpha}\right)\left[1+\left(\frac{x-\lambda}{\alpha}\right)\right]^{-2} \tag{1}$$

Scale, shape and origin were all represented by the aforementioned parameters $\alpha$, $\beta$, and $\gamma$, respectively. The probability distribution function can, therefore, be expressed as follows.

$$F(x) = \left[1+\left(\frac{\alpha}{x-\gamma}\right)^{\beta}\right]^{-1} \tag{2}$$

The SPEI may be simply calculated using the standardized values of *F(x)*

$$\text{SPEI} = W - \frac{C_0 + C_1 W + C_2 W^2}{1 + d_1 W + d_2 W^2 + d_3 W^3} \tag{3}$$

when $P \leq 0.5$, $W = \sqrt{-2ln(P)}$, and when $P > 0.5$, $W = \sqrt{-2ln(1-P)}$, $C_0 = 2.5155$, $C_1 = 0.8028$, $C_2 = 0.0203$, $d_1 = 1.4327$, $d_2 = 0.1892$, and $d_3 = 0.0013$.

The drought index was calculated using the SPEI package in the *R* environment [29]. Negative SPEI values, accompanied by a decrease in rainfall, signal drought, whereas positive SPEI values signal wetter conditions.

### 2.3.2. Machine Learning Algorithms

For the perdition multiscale SPEI, this study considered three tree-based algorithms, random subspace (RSS), additive regression (AR), and M5 pruned (M5P), and their stacking hybrid forms, AR-RSS, AR-M5P, RSS-M5P, and AR-RSS-M5P. The basic concept of tree-based algorithms is fitting decision trees to different sub-samples of calibration datasets and then integrating the prediction of each tree to provide the final output. Fitting models to different sub-samples allows data decomposition in detail for better identification of complex input–output relations. The advantages of these non-parametric methods are accurate prediction, little data pre-processing such as normalization or scaling, and no assumptions on space distributions. A detailed description of the standalone model can be found in the study by Elbeltagi et al. [68]. Wolpert [69] proposed the use of a hybrid stacking algorithm. This method facilitates the use of ensemble algorithms, which combine two or more algorithms throughout the training period. The idea behind stacking hybrid generalization is to use first-level learners to train and predict training datasets. For the meta learner, a new training dataset was created by combining the predicted outcomes from first-level learners. Sikora et al. [70] provided comprehensive information on stacked hybrid generalization.

All modelings were carried out using the WEKA (Version3.8.4, Waikato University, Hamilton, New Zealand) software (https://www.cs.waikato.ac.nz/ml/weka, accessed on 20 May 2021). Table 2 contains the model parameters. Additionally, a flowchart illustrating the methodology used is shown in Figure 2. The WEKA software is a collection of ML techniques related to data mining. It can manage and process various data types using multiple tools, i.e., regression, clustering, and visualization. The program also features a graphi-

cal user interface (GUI), which facilitates the program's operation. There are additional alternatives, such as MATLAB, Python, and R, but preparing and implementing computer codes using those programs requires considerable time. However, the implementation of models in different environments does not affect the obtained results. Therefore, WEKA was chosen considering its simple implementation procedure compared to other software.

**Table 2.** The parameters of the machine learning algorithm in modeling SPEI-6, -9, -12 and -24.

| Model Names | Parameter Descriptions |
| --- | --- |
| Random Subspace (RSS) | Batch size-100, Classifier = REPTree, random seed-1, subspace size = 0.5, numbers of executions slots = 1, number of iterations = 10 |
| Additive Regression (AR) | Batch size-100, Classifier = Bagging, shrinkage = 1, number of iterations = 30 |
| M5 Pruned (M5P) | Batch size-100, Minimum number of instances = 4 |

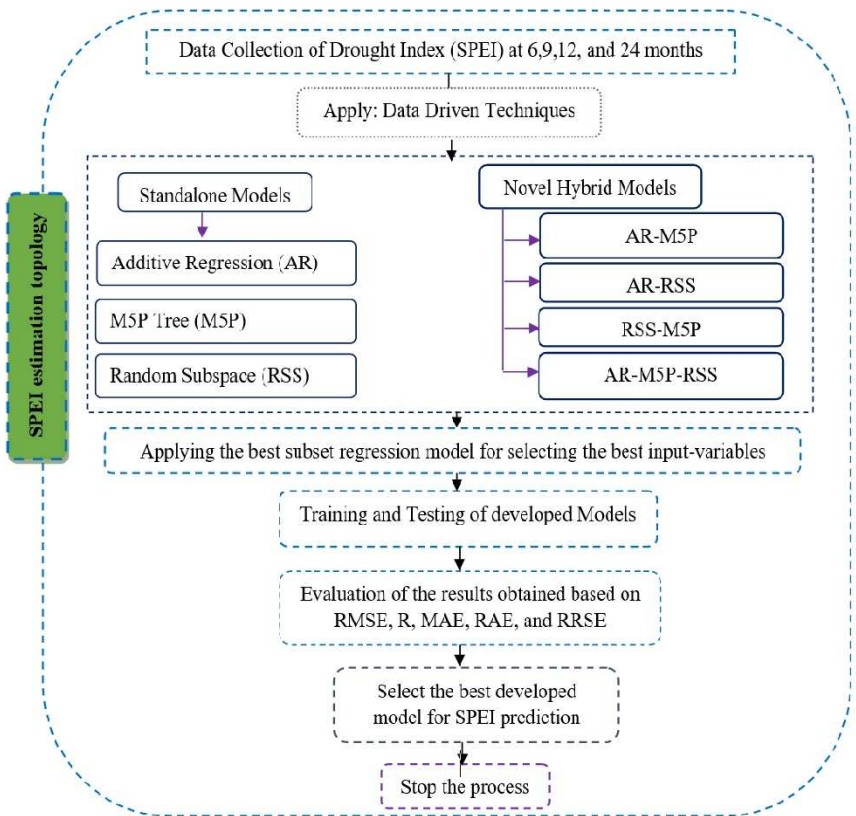

**Figure 2.** Flowchart of SPEI estimation methodology in the study area.

A grid search optimization algorithm was used to select optimum ML model parameters. It randomly searches the optimum parameters within a discrete grid space. The value within the range that provides the most accurate prediction is selected. Table 2 shows the selected optimum ML model hyperparameter values.

### 2.4. Constructing and Evaluating Models

Choosing suitable input and output variables for non-linear hydrologic systems can be time-consuming. SPEI 6, 9, 12, and 24 were calculated using precipitation and temperature data from Rajshahi station. The most important inputs (at different time lags) for the target variable (output) were selected by using subset regression and sensitivity analyses. When there are many input variables, one of the most critical stages in the soft computing model is feature selection. Numerous procedures, such as best subset regression, forward stepwise

selection, and mutual information, can be used to determine the best combinations of all possible inputs. The current study employed best subset regression to determine the best possible input combinations for the SPI 6, 9-, 12-, and 24-month models. The optimal input combination was determined using six statistical criteria (MSE, determination coefficients (R2), adjusted R2, Akaike's AIC, Mallows' Cp, Akaike's AIC, and Amemiya's PC).

Multiscale SPEI data were predicted using standalone ML models and their stacking hybrid forms at the Rajshahi station. The models were developed using 30-year monthly data (1980–2009), and 9-year data (2010–2018) were used to test the model at Rajshahi station. Evaluation of model outputs was carried out using statistical indices and visual interpretation, including Taylor diagram, scatter, and boxplots. The models with the lowest RMSE, MAE, RAE, RRSE, and greater R during testing were considered superior for drought prediction. An evaluation was performed on the generalizability of the most successful model developed at Rajshahi station to predict SPEI at other stations (Bogra, Mymensingh, Khulna and Rangpur).

### 2.5. Statistical Performance Measurement

Several performance metrics were calculated to assess the model performance, including the root mean square error (*RMSE*), coefficient of determination (*R*), mean absolute error (*MAE*), relative absolute error (*RAE*), and root relative squared error (*RRSE*). Definitions for all parameters are as follows:

$$RMSE = \sqrt{\frac{\sum_{i=1}^{N}\left(\text{SPEI}_{Obs} - \text{SPEI}_{pre}\right)^2}{N}} \tag{4}$$

$$MAE = \frac{\sum_{i=1}^{N}\left|\left(\text{SPEI}_{Obs} - \text{SPEI}_{pre}\right)\right|}{N} \tag{5}$$

$$RAE = \left|\frac{\text{SPEI}_{Obs} - SPEI_{pre}}{\text{SPEI}_{pre}}\right| \times 100 \tag{6}$$

$$RRSE = \frac{\sqrt{\sum_{i=1}^{N}\left(\text{SPEI}_{pre} - \text{SPEI}_{Obs}\right)^2}}{\sqrt{\sum_{i=1}^{N}\left(\text{SPEI}_{Obs} - \text{SPEI}_{pre}\right)^2}} \tag{7}$$

$$R = \frac{\sum_{i=1}^{N}\left(\text{SPEI}_{Obs} - \overline{\text{SPEI}_{Obs}}\right)\left(\text{SPEI}_{pre} - \overline{\text{SPEI}_{pre}}\right)}{\sqrt{\sum_{i=1}^{N}\left(\text{SPEI}_{Obs} - \overline{\text{SPEI}_{Obs}}\right)^2}\sqrt{\sum_{i=1}^{N}\left(\text{SPEI}_{pre} - \overline{\text{SPEI}_{pre}}\right)^2}} \tag{8}$$

The actual and predicted SPEI*s* are represented by SPEI*_{obs}* and SPEI*_{pre}*, respectively, with $\overline{\text{SPEI}}$ representing the average values of the actual SPEI index, and *N* representing the number of observations.

## 3. Results

### 3.1. Model Input Selection

Table 3 summarizes the statistical indices of model performance for various input combinations. Best results are indicated by bolded numbers in Table 3. The results indicate that three lag values, 1, 6, and 7, were the most accurate predictors of SPI-6. Similarly, inputs 1, 2, 3, 4, 6, 7, and 8 provided the best forecasts for SPEI-9 and SPEI-12, but only lag 1 did so for SPEI-24.

The input variables were analyzed for sensitivity at a 5% significance level. The results from the regression analysis for SPEI-6, SPEI-9, SPEI-12, and SPEI-24 are depicted in Figure 3. Spikes with lags of 1, 6, 7, 8 for SPEI-6; 1, 2, 3, 4, 6, 7, 8 for SPEI-9; 1, 3, 4, 5, 6, 7, 8 for SPEI-12, and 1 for SPEI-24 were statistically significant, according to the findings. Sensitivity analysis verified the results of the statistical metrics presented in Table 3. In this way, the SPEI of the corresponding scale was predicted using those lags. The inputs used

to develop the SPEI for various temporal scales at the Rajshahi station are summarized in Table 4.

**Table 3.** The best subset regression analysis for determining the best input combinations to model.

| Variables | MSE | $R^2$ | Adjusted $R^2$ | Mallows' Cp | Akaike's AIC | Amemiya's PC |
|---|---|---|---|---|---|---|
| **SPEI-6** | | | | | | |
| SPEI1 | 0.404 | 0.581 | 0.580 | 31.480 | −417.726 | 0.421 |
| SPEI1/SPEI6 | 0.396 | 0.590 | 0.588 | 22.969 | −425.756 | 0.413 |
| **SPEI1/SPEI6/SPEI7** | **0.379** | **0.609** | **0.606** | **3.166** | **−445.282** | **0.396** |
| SPEI1/SPEI3/SPEI6/SPEI7 | 0.377 | 0.611 | 0.608 | 2.119 | −446.371 | 0.395 |
| SPEI1/SPEI3/SPEI4/SPEI6/SPEI7 | 0.378 | 0.612 | 0.607 | 3.785 | −444.711 | 0.397 |
| SPEI1/SPEI3/SPEI4/SPEI5/SPEI6/SPEI7 | 0.378 | 0.612 | 0.607 | 5.141 | −443.367 | 0.398 |
| SPEI1/SPEI3/SPEI4/SPEI5/SPEI6/SPEI7/SPEI8 | 0.379 | 0.612 | 0.606 | 7.072 | −441.437 | 0.400 |
| SPEI1/SPEI2/SPEI3/SPEI4/SPEI5/SPEI6/SPEI7/SPEI8 | 0.380 | 0.612 | 0.605 | 9.000 | −439.511 | 0.401 |
| **SPEI-9** | | | | | | |
| SPEI1 | 0.246 | 0.745 | 0.744 | 11.701 | −642.458 | 0.256 |
| SPEI1/SPEI8 | 0.243 | 0.749 | 0.748 | 6.194 | −647.901 | 0.253 |
| SPEI1/SPEI3/SPEI8 | 0.241 | 0.752 | 0.750 | 2.947 | −651.176 | 0.251 |
| SPEI1/SPEI3/SPEI4/SPEI8 | 0.240 | 0.753 | 0.750 | 3.533 | −650.608 | 0.252 |
| SPEI1/SPEI3/SPEI4/SPEI7/SPEI8 | 0.240 | 0.753 | 0.751 | 4.020 | −650.145 | 0.252 |
| SPEI1/SPEI2/SPEI3/SPEI4/SPEI7/SPEI8 | 0.240 | 0.754 | 0.750 | 5.394 | −648.782 | 0.253 |
| **SPEI1/SPEI2/SPEI3/SPEI4/SPEI6/SPEI7/SPEI8** | **0.241** | **0.754** | **0.750** | **7.016** | **−647.168** | **0.254** |
| SPEI1/SPEI2/SPEI3/SPEI4/SPEI5/SPEI6/SPEI7/SPEI8 | 0.241 | 0.754 | 0.750 | 9.000 | −645.184 | 0.255 |
| **SPEI-12** | | | | | | |
| SPEI1 | 0.151 | 0.844 | 0.844 | 6.523 | −861.581 | 0.156 |
| SPEI1/SPEI8 | 0.148 | 0.848 | 0.847 | −0.607 | −868.752 | 0.154 |
| SPEI1/SPEI5/SPEI8 | 0.149 | 0.848 | 0.847 | 0.857 | −867.297 | 0.154 |
| SPEI1/SPEI6/SPEI7/SPEI8 | 0.149 | 0.848 | 0.847 | 2.160 | −866.005 | 0.155 |
| SPEI1/SPEI3/SPEI5/SPEI7/SPEI8 | 0.149 | 0.848 | 0.846 | 3.684 | −864.489 | 0.155 |
| SPEI1/SPEI3/SPEI4/SPEI6/SPEI7/SPEI8 | 0.149 | 0.848 | 0.846 | 5.127 | −863.057 | 0.156 |
| **SPEI1/SPEI3/SPEI4/SPEI5/SPEI6/SPEI7/SPEI8** | **0.149** | **0.848** | **0.846** | **7.023** | **−861.164** | **0.156** |
| SPEI1/SPEI2/SPEI3/SPEI4/SPEI5/SPEI6/SPEI7/SPEI8 | 0.150 | 0.848 | 0.846 | 9.000 | −859.187 | 0.157 |
| **SPEI-24** | | | | | | |
| **SPEI1** | **0.081** | **0.916** | **0.916** | **1.071** | **−1116.440** | **0.084** |
| SPEI1/SPEI5 | 0.081 | 0.917 | 0.916 | 1.240 | −1116.287 | 0.084 |
| SPEI1/SPEI5/SPEI7 | 0.081 | 0.917 | 0.917 | 0.835 | −1116.724 | 0.084 |
| SPEI1/SPEI5/SPEI7/SPEI8 | 0.081 | 0.918 | 0.917 | 1.403 | −1116.182 | 0.084 |
| SPEI1/SPEI5/SPEI6/SPEI7/SPEI8 | 0.081 | 0.918 | 0.917 | 3.057 | −1114.535 | 0.084 |
| SPEI1/SPEI2/SPEI5/SPEI6/SPEI7/SPEI8 | 0.081 | 0.918 | 0.917 | 5.033 | −1112.559 | 0.085 |
| SPEI1/SPEI2/SPEI4/SPEI5/SPEI6/SPEI7/SPEI8 | 0.081 | 0.918 | 0.916 | 7.025 | −1110.568 | 0.085 |
| SPEI1/SPEI2/SPEI3/SPEI4/SPEI5/SPEI6/SPEI7/SPEI8 | 0.081 | 0.918 | 0.916 | 9.000 | −1108.593 | 0.085 |

SPI1 = SPI (t-1), SPI8 = SPI (t-8); Bold: indicates the selected input combinations.

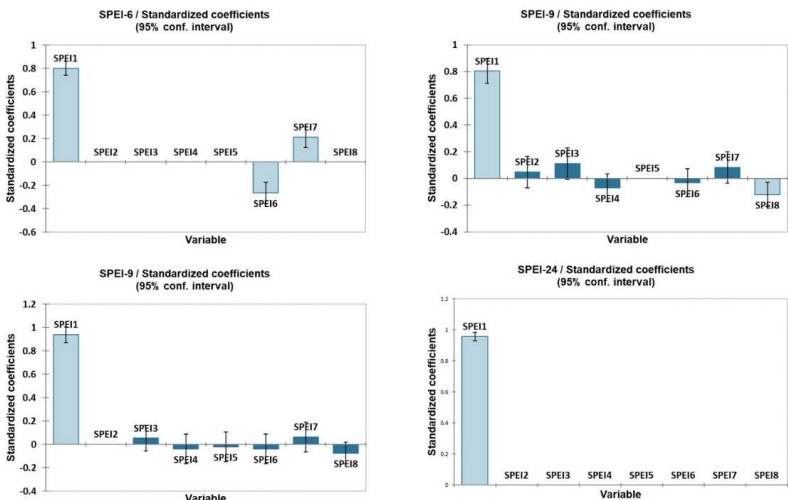

**Figure 3.** The standardized coefficients of input variables for sensitivity analysis (SPEI-6, 9, 12, and 24).

**Table 4.** Input variables selected for multiscale SPEI prediction.

| Output | Input Variables |
|---|---|
| SPEI6 | SPEI (t-1), SPEI (t-6), SPEI (t-7) |
| SPEI9 | SPEI (t-1), SPEI (t-2), SPEI (t-3), SPEI (t-4), SPEI (t-6), SPEI (t-7), SPEI (t-8) |
| SPEI12 | SPEI (t-1), SPEI (t-3), SPEI (t-4), SPEI (t-5), SPEI (t-6), SPEI (t-7), SPEI (t-8) |
| SPEI24 | SPEI (t-1) |

### 3.2. Prediction of Droughts Using Machine Learning Techniques

To forecast multiscale SPEI data at Rajshahi station, individual and hybrid model versions were developed. It took 30 years of data from 1980 to 2009 to create models and data from 2010 to 2018 for testing at Rajshahi station. Various statistical indices and visual interpretations were used to evaluate the models' performance, including Taylor diagrams, scatter diagrams, and boxplots. As a general rule, models with a lower RMSE, MAE, RAE, RRSE, and a larger R during testing were deemed more accurate for drought prediction. According to statistical metrics, the models' performance at Rajshahi station is shown in Table 5. The best results are shown in the table in bold type.

Based on the testing data, the best-performing model was selected. Table 5 shows that M5P was the best predictor of SPEI on all time scales, with the highest accuracy. M5P had the lowest RRSE, RAE, MAE and RMSE, and the highest $R$ of any other model when predicting SPEI over various time scales, with $R$ ranging from 0.750 to 0.98, MAE ranging from 0.208 to 0.49, and the RMSE from 0.389 to 0.67; RAE was between 16.60% and 55.26%, and RRSE was between 27.89% and 62.92%. There was a positive correlation between increasing SPEI time scales and improved model accuracy. Consequently, SPEI-24 had the best prediction accuracy.

Taylor diagrams (TD) were used to examine the geographic configuration of predicted and calculated (observed) multiscale SPEI values based on various ML models during testing. Model performance can be evaluated visually using Taylor [64], which provides a polar plot to show how models reproduce observed values while emphasizing their accuracy and precision. TD shows three statistical matrices, including standard deviation (SD), root mean square error (RMSE), and correlation coefficient ($R$). There was a good correlation between the Taylor diagram (Figure 4) and the derived performance indicators in Table 4. SPEI has the best M5P prediction–observation agreement (orange triangle) of any timeframe except SPEI9, as shown by the Taylor diagram. The lowest RMSE (0.424) and highest correlations (0.869) were for the M5P model for SPEI9, while the variation in RSS-M5P (0.968) was the most in line with observations.

**Table 5.** Computed statistical index values for the eight machine learning models (individual and hybrid) during training and testing stages.

| Models | Training Period (1980–2009) | | | | | Testing Period (2010–2018) | | | | |
|---|---|---|---|---|---|---|---|---|---|---|
| | *R* | *MAE* | *RMSE* | *RAE (%)* | *RRSE (%)* | *R* | *MAE* | *RMSE* | *RAE (%)* | *RRSE (%)* |
| | | | | | **SPEI6** | | | | | |
| AR | 0.929 | 0.268 | 0.363 | 34.20 | 37.70 | 0.675 | 0.594 | 0.770 | 66.90 | 71.90 |
| M5P | 0.785 | 0.443 | 0.596 | 56.58 | 61.97 | **0.757** | **0.491** | **0.674** | **55.26** | **62.92** |
| RSS | 0.809 | 0.431 | 0.567 | 54.99 | 58.95 | 0.687 | 0.571 | 0.740 | 64.32 | 69.12 |
| AR-M5P | 0.786 | 0.453 | 0.597 | 57.83 | 62.06 | 0.707 | 0.542 | 0.715 | 60.94 | 66.81 |
| AR-RSS | 0.788 | 0.458 | 0.593 | 58.46 | 61.66 | 0.642 | 0.597 | 0.770 | 67.20 | 71.97 |
| RSS-M5P | 0.7947 | 0.444 | 0.586 | 56.72 | 61.02 | 0.713 | 0.536 | 0.708 | 60.29 | 66.16 |
| AR-M5P-RSS | 0.7697 | 0.473 | 0.618 | 60.37 | 64.27 | 0.697 | 0.547 | 0.722 | 61.54 | 67.43 |

Numbers in boldface indicate the ideal values.

**Table 5.** *Cont.*

| Models | Training Period (1980–2009) | | | | | Testing Period (2010–2018) | | | | |
|---|---|---|---|---|---|---|---|---|---|---|
| | *R* | *MAE* | *RMSE* | *RAE (%)* | *RRSE (%)* | *R* | *MAE* | *RMSE* | *RAE (%)* | *RRSE (%)* |
| **SPEI9** | | | | | | | | | | |
| AR | 0.968 | 0.168 | 0.246 | 21.35 | 25.25 | 0.747 | 0.444 | 0.600 | 53.97 | 57.32 |
| M5P | 0.879 | 0.323 | 0.465 | 41.03 | 47.61 | **0.763** | **0.397** | **0.571** | **48.31** | **54.53** |
| RSS | 0.893 | 0.316 | 0.444 | 40.14 | 45.48 | 0.762 | 0.426 | 0.573 | 51.77 | 54.76 |
| AR-M5P | 0.858 | 0.346 | 0.503 | 44.01 | 51.55 | 0.736 | 0.447 | 0.615 | 54.34 | 58.75 |
| AR-RSS | 0.922 | 0.284 | 0.387 | 36.14 | 39.65 | 0.718 | 0.460 | 0.619 | 55.95 | 59.08 |
| RSS-M5P | 0.882 | 0.320 | 0.461 | 40.70 | 47.23 | 0.742 | 0.438 | 0.612 | 53.23 | 58.44 |
| AR-M5P-RSS | 0.859 | 0.345 | 0.502 | 43.91 | 51.42 | 0.740 | 0.460 | 0.619 | 55.95 | 59.08 |
| **SPEI12** | | | | | | | | | | |
| AR | 0.977 | 0.133 | 0.210 | 16.95 | 21.45 | 0.833 | 0.339 | 0.483 | 40.62 | 45.21 |
| M5P | 0.925 | 0.254 | 0.373 | 32.33 | 38.06 | **0.869** | **0.281** | **0.424** | **33.56** | **39.70** |
| RSS | 0.909 | 0.321 | 0.434 | 40.74 | 44.26 | 0.711 | 0.472 | 0.617 | 56.43 | 57.82 |
| AR-M5P | 0.926 | 0.249 | 0.370 | 31.67 | 37.80 | 0.851 | 0.306 | 0.44 | 36.62 | 41.62 |
| AR-RSS | 0.954 | 0.209 | 0.298 | 26.55 | 30.41 | 0.802 | 0.407 | 0.53 | 48.64 | 49.42 |
| RSS-M5P | 0.934 | 0.243 | 0.350 | 30.86 | 35.71 | 0.849 | 0.313 | 0.45 | 37.44 | 42.20 |
| AR-M5P-RSS | 0.928 | 0.246 | 0.365 | 31.21 | 37.22 | 0.847 | 0.324 | 0.456 | 38.74 | 42.75 |
| **SPEI24** | | | | | | | | | | |
| AR | 0.975 | 0.140 | 0.197 | 19.28 | 22.19 | 0.908 | 0.227 | 0.403 | 18.18 | 28.92 |
| M5P | 0.960 | 0.168 | 0.247 | 23.09 | 27.90 | **0.928** | **0.208** | **0.389** | **16.60** | **27.89** |
| RSS | 0.969 | 0.156 | 0.221 | 21.47 | 24.91 | 0.915 | 0.232 | 0.401 | 18.54 | 28.75 |
| AR-M5P | 0.958 | 0.182 | 0.254 | 25.09 | 28.71 | 0.884 | 0.269 | 0.436 | 21.49 | 31.30 |
| AR-RSS | 0.959 | 0.183 | 0.251 | 25.13 | 28.38 | 0.878 | 0.280 | 0.441 | 22.37 | 31.62 |
| RSS-M5P | 0.963 | 0.172 | 0.241 | 23.70 | 27.16 | 0.874 | 0.283 | 0.440 | 22.68 | 31.58 |
| AR-M5P-RSS | 0.962 | 0.173 | 0.242 | 23.84 | 27.32 | 0.886 | 0.270 | 0.433 | 21.59 | 31.09 |

Numbers in boldface indicate the ideal values.

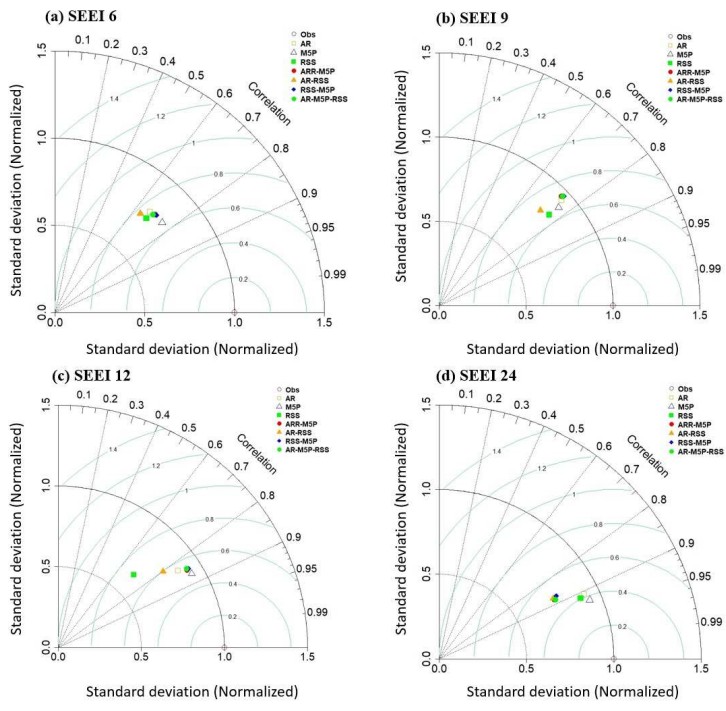

**Figure 4.** Taylor diagram representation of model's performance in predicting SPEIs at Rajshahi station over multiple time scales during the testing period (**a**) SEEI-6; (**b**) SEEI-9; (**c**) SEEI-12; (**d**) SEEI-24.

During the testing period (2010–2018), model performance at Rajshahi station was evaluated using scatter plots (Figure 5). Predicted and observed values showed good correlation. The ideal line (a 45° line) was aligned with the majority of the predicted points, indicating that all models had a high degree of accuracy in their predictions. For all timeframes, M5P had the best correlation coefficient performance ($R = 0.76–0.93$), while AR-RSS had the worst ($R = 0.64–0.87$). On average, the scatter plots indicated that the M5P outperformed the other models.

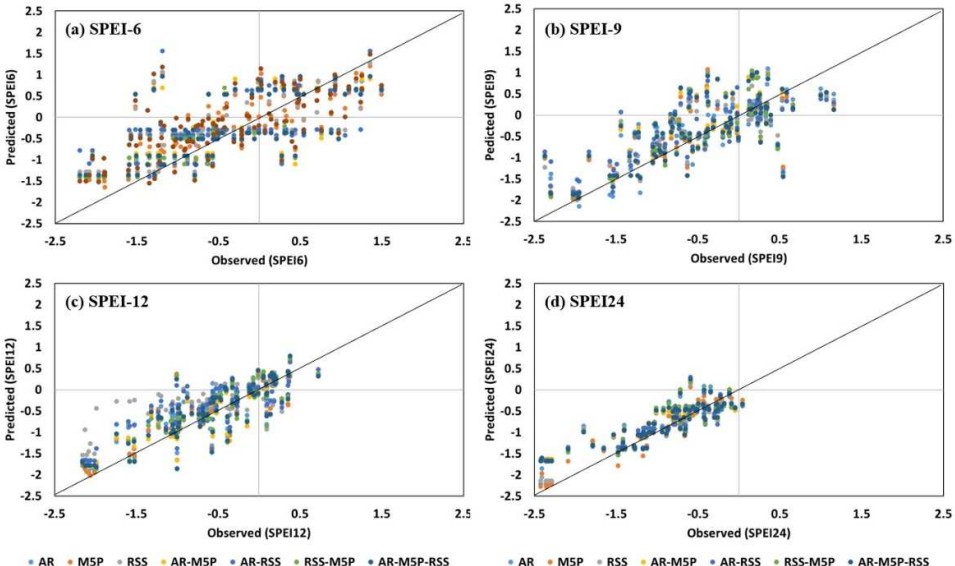

**Figure 5.** The scatter plots depict the observed and predicted SPEI values for various models at Rajshahi station over multiple time scales during the testing period, (**a**) SPEI-6, (**b**) SPEI-9, (**c**) SPEI-12, (**d**) SPEI-24.

SPEI data were used to create boxplots showing the 25%, 50%, and 75% quantiles of the observed and predicted SPEI values (Figure 6). All prediction models performed admirably in predicting SPEI quantiles at various scales, especially at higher orders. In simulating SPEI quantiles at all scales, the M5P model showed the best overall accuracy, while the RSS and AR models performed the worst.

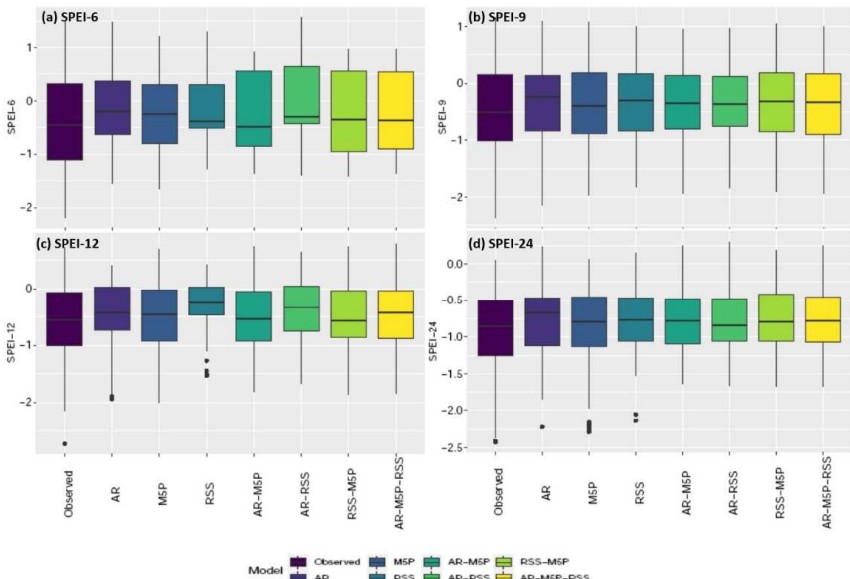

**Figure 6.** Predictive model performance at Rajshahi station over multiple time scales during the testing period, (**a**) SPEI-6, (**b**) SPEI-9, (**c**) SPEI-12, (**d**) SPEI-24.

### 3.3. The Best Predictive Model Is Applied in Various Regions

The best predictive model (M5P) was used to predict multi-scaler SPEI values at four different stations located throughout the drought-prone northern part of Bangladesh: Rangpur, Borga, Mymensingh, and Khulna. Model performance was evaluated at the test locations for the whole study period (1980–2018). According to various statistical indices, the performance evaluation results are summarized in Table 6. The model performed satisfactorily in terms of the statistical indices computed. The M5P predicted the SPEI-6, 9, 12, and 24 with *R* values ranging from 0.787 to 0.802, 0.850 to 0.882, 0.899 to 0.938, and 0.927 to 0.966, respectively. The MAE, RMSE, RAE, and RRSE were all low in terms of prediction on all time scales. In terms of accuracy, SPEI-24 was predicted the best by the M5P model, followed by SPEI-9 and SPEI-6. According to the findings, drought forecasting in western Bangladesh could benefit from the M5P model developed in this study.

**Table 6.** Statistical performance of M5P model in predicting SPEIs at the test stations.

| Models | Statistical Indices | | | | |
|---|---|---|---|---|---|
| | *R* | *MAE* | *RMSE* | *RAE* | *RRSE* |
| SEPI6-Bogra | 0.802 | 0.44 | 0.58 | 54.50 | 59.47 |
| SEPI6-Khulna | 0.793 | 0.45 | 0.60 | 54.14 | 60.65 |
| SEPI6-Mymensingh | 0.787 | 0.46 | 0.61 | 55.37 | 61.37 |
| SEPI6-Rangpur | 0.791 | 0.46 | 0.61 | 56.09 | 60.85 |
| SEPI9-Bogra | 0.882 | 0.31 | 0.46 | 38.03 | 46.65 |
| SEPI9-Khulna | 0.850 | 0.37 | 0.52 | 43.58 | 52.32 |
| SEPI9-Mymensingh | 0.880 | 0.33 | 0.47 | 40.81 | 46.99 |
| SEPI9-Rangpur | 0.861 | 0.35 | 0.51 | 42.79 | 50.38 |
| SEPI12-Bogra | 0.938 | 0.24 | 0.34 | 29.04 | 34.34 |
| SEPI12-Khulna | 0.899 | 0.30 | 0.44 | 36.15 | 43.37 |
| SEPI12-Mymensingh | 0.925 | 0.26 | 0.37 | 31.57 | 37.43 |
| SEPI12-Rangpur | 0.929 | 0.24 | 0.37 | 30.13 | 36.65 |
| SEPI24-Bogra | 0.962 | 0.17 | 0.27 | 20.60 | 26.45 |
| SEPI24-Khulna | 0.927 | 0.24 | 0.37 | 28.25 | 36.17 |
| SEPI24-Mymensingh | 0.961 | 0.18 | 0.27 | 21.94 | 26.80 |
| SEPI24-Rangpur | 0.966 | 0.16 | 0.26 | 19.28 | 25.17 |

## 4. Discussion

Droughts frequently affect agriculture and the livelihoods of farmers in northern Bangladesh. Reliable forecasting of droughts is important to reduce drought impacts in the region. However, drought analysis and prediction need reliable rainfall and temperature data recorded for longer periods, which most of the meteorological stations in Bangladesh do not have. ML algorithms were used in this study to overcome the limitations of climate data. The present study revealed that the SPEI, the most widely used DI, can be accurately forecasted using ML models for a multi-month horizon (i.e., 6, 9, 12 and 24). SPEI6, SPEI9, SPEI12, and SPEI24-month models were optimized using the best subset regression analysis. Predictions of SPEI using M5P were the most accurate across all time scales. The temporal variability of SPEIs could be replicated in different parts of western Bangladesh. Water professionals and policymakers may find this model useful for making intelligent decisions.

The studied models better predicted SPEIs over longer periods than shorter periods. This is expected as the smoothness and randomness of SPEI time series increase with increasing time scale. More linear data improve the machine learning models' performance, as was found in the current study. M5P was still able to predict lower-scale SPEI series with an *R* = 0.76. A highly non-linear process can be captured using the M5P model.

A multivariate linear algorithm is how the M5P model works. The leaves on the tree represent different linear regression models. Segmenting data and fitting it with an appropriate regression model are made easier with this method [71]. Due to its decomposition capability, it can fit different models to various non-linear datasets. M5P's ability to

simulate each data point in a data series helped it to better predict linear model phenomena than the other models tested in this study. M5P's ability to learn efficiently and model high-dimensional data also improved as a result of this change.

Several drought indicators have been predicted using algorithms inspired by nature and a stochastic (time-series) model. The M5P results were compared to those of these models (DIs). The AR and non-linear bi-linear (BL) models based on global climate indicators and delayed SPI data were used to predict the meteorological drought in Ankara, Turkey [45]. The predictive power of the BL and AR models was lower than the prediction accuracy obtained in this study. The least-squares support vector machine (LSSVM), multivariate adaptive regression splines (MARS), and M5 tree models were used to predict droughts in eastern Australia [42]. Prediction accuracy was reported to be higher for the M5 tree approach. Özger et al. [72] used ANN and SVM and their hybrids with wavelet decomposition for forecasting droughts in the Antalya region of Turkey. They showed that hybridization improved model performance significantly, which agrees with the present study's findings. Nguyen et al. [73] used different ML models, including ANFIS, M5, M11 and M13 models, to predict SPEI in the Cai River basin of Vietnam. They reported M5 as the best performing model, followed by M11 and M13. Adarsh and Janga Reddy [74] employed stepwise linear regression, genetic programming, and M5 methods to predict standardized precipitation indices for different regions of India. They reported superior performance from M5 in predicting droughts in all regions. Shamshirband et al. [75] predicted SPEI using support vector regression, gene expression programming (GEP), and M5 models and reported higher performance with the M5 model. Barzkar et al. [76] used three ML models, GEP, M5, and multivariate adaptive regression spline (MARS), to predict SPEIs for different climatic conditions. They showed that the M5 model performed better in all cases.

The literature mentioned above clearly indicates the ML models' capability to predict droughts in different meteorological settings. The present study revealed that the ML model, specifically M5P, was more capable of forecasting meteorological droughts over a wide range of timescales. The study revealed that a hybrid ML model can significantly improve the standalone ML models' performance. Longer-period droughts are more predictable than shorter-period droughts. This may be due to the higher variability of shorter-period droughts. Droughts have a devastating impact on society and the economy. The findings of this study indicate that drought forecasting models in drought-prone eastern regions of Bangladesh have the potential to be installed as an early warning system to mitigate the effects of drought.

## 5. Conclusions

Newly developed machine learning models for forecasting SPEI in Bangladesh were evaluated in this study. In Rajshahi, Bangladesh's most drought-prone location, the models were developed to predict SPEI for the period from 1980 to 2018. The model was validated at four stations distributed over the country's western region. SPEI was accurately predicted in Rajshahi using the M5P model. $R$ values for SPEI-6, 9, 12, and 24 at the validation stations were 0.787–0.802, 0.850–0.882, 0.899–0.938 and 0.927–0.966, respectively. The M5P model can forecast droughts on multiple timescales according to correlation and low errors. As a result of climate change, the model's output could help predict when droughts will occur in the country and help mitigate their growing negative effects. This study considered only three tree-based algorithms for drought prediction. Bangladesh's droughts can be predicted using other ML models to see how well they work. Optimization algorithms can be combined with machine learning models to improve predictability.

**Author Contributions:** Each author contributed to the composition of the paper. S.S., F.A. and M.K. supervised and coordinated this study; A.E. performed the calculations and modeling; D.K.R., P.K.K., L.D., M.M.I. and M.M.R. assisted in reviewing the manuscript. All authors have read and agreed to the published version of the manuscript.

**Funding:** There was no external funding for this research.

**Institutional Review Board Statement:** Not applicable.

**Informed Consent Statement:** Not applicable.

**Data Availability Statement:** No applicable.

**Conflicts of Interest:** The authors declare no conflict of interest.

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
