# Peer review of "Estimating the Standardized Precipitation Evapotranspiration Index Using Data-Driven Techniques: A Regional Study of Bangladesh"

_water, doi:10.3390/w14111764_

Round 1
Reviewer 1 Report
1. The contribution is not stated clearly.
2. The choice of parameters used in the algorithm is not well justified.
3. A deep and detailed comparison with other methods is mandatory.
4. The authors claim that their method is faster and more efficient, but this is not rigorously demonstrated since it is applied just for a particular case.
5. What do you mean by experimental validation? Where the data exactly comes from, what is their reliability and accuracy for which model? Please address this important point seriously. Authors must cite the following papers;
Overall the quality of this paper is very good.
I recommend this paper. Authors must cite the following papers:
Kumar, R., & Dhiman, G. (2021). A Comparative Study of Fuzzy Optimization through Fuzzy Number. International Journal of Modern Research, 1, 1-14.
Chatterjee, I. (2021). Artificial Intelligence and Patentability: Review and Discussions. International Journal of Modern Research, 1, 15-21.
Vaishnav, P.K., Sharma, S., & Sharma, P. (2021). Analytical Review Analysis for Screening COVID-19. International Journal of Modern Research, 1, 22-29.
Author Response
Point 1: The contribution is not stated clearly.
Response 1: The contribution is developing novel hybrid artificial intelligence models for estimating SPEI. We added two new paragraphs at the end of the Introduction section to discuss the contribution of the study clearly. We clearly mentioned that no research has been published in the literature to the best of our knowledge that predicts the SPEI in the study region using ML models.
Point 2:The choice of parameters used in the algorithm is not well justified.
Response 2: We used a grid search algorithm to select the optimum values of model parameters. We discussed it in the revised manuscript as below:
“A grid search optimization algorithm was used to select optimum ML model parameters. It randomly searches the optimum parameters within a discrete grid space. The value within the range that provides the most accurate prediction is selected. Table 2 shows the selected optimum ML model hyperparameter values.”
Point3: A deep and detailed comparison with other methods is mandatory.
Response 3: We mentioned in the introduction of the paper that similar studies have not been done before in Bangladesh. Therefore, it was not possible to compare the performance with other studies or models used for drought prediction. However, we have compared the results of our study with the performance of ML models in drought predictions in recent literature in different regions. New tests have been added in the Discussion section of the revised manuscript for this purpose. The newly added texts are as below:
“Several drought indicators have been predicted using algorithms inspired by nature and a stochastic (time-series) model. The M5P results were compared to those of these models (DIs). The AR and non-linear bi-linear (BL) models based on global climate indicators and delayed SPI data were used to predict the meteorological drought in Ankara, Turkey [53]. The predictive power of the BL and AR models was less than the prediction accuracy obtained in this study. The least-square support vector machine (LSSVM), multivariate adaptive regression splines (MARS), and M5 tree models were used to predict droughts in eastern Australia[50]. Prediction accuracy was reported higher for the M5 tree approach. . Özger et al. [74] ANN and SVM, and their hybridization with wavelet decomposition for forecasting droughts in the Antalya region of Turkey. They showed hybridization im-proved the model performance significantly, which collaborates with the present study's findings. Nguyen et al. [75] used different ML models, including ANFIS, M5, M11 and M13 models, to predict SPEI in the Cai River basin of Vietnam. They reported M5 as the best performing model, followed by M11 and M13. Adarsh and Janga Reddy [76] em-ployed stepwise linear regression, genetic programming, and M5 methods for predicting standardized precipitation index in different regions of India. They reported superior performance M5 in predicting droughts in all regions. Shamshirband et al. [77]predicted SPEI using Support Vector Regression, Gene Expression Programming (GEP), and M5 models, and reported higher performance of M5 model. Barzkar et al. [78] used three ML models: GEP, M5, and Multivariate Adaptive Regression Spline (MARS) in predicting SPEI in different climatic conditions. They showed that the M5P model performed better in all cases.”
Point 4: The authors claim that their method is faster and more efficient, but this is not rigorously demonstrated since it is applied just for a particular case.
Response 4: We removed the statement. However, we showed the capability of the models used in this study to predict droughts in Bangladesh. It is not possible to claim that the selected models are the most efficient among numerous available ML algorithms and their hybrid versions. Therefore, we mentioned the following as further recommended work at the end of the Conclusion section:
“Bangladesh’s droughts can be predicted using other ML models to see how well they work. Optimization algorithms can be combined with machine learning models to improve predictability.”
Point 5: What do you mean by experimental validation? Where the data exactly come from, and what is their reliability and accuracy for which model? Please address this important point seriously. Authors must cite the following papers.
Overall, the quality of this paper is very good.
I recommend this paper. Authors must cite the following papers:
Kumar, R., & Dhiman, G. (2021). A Comparative Study of Fuzzy Optimization through Fuzzy Number. International Journal of Modern Research, 1, 1-14.
Chatterjee, I. (2021). Artificial Intelligence and Patentability: Review and Discussions. International Journal of Modern Research, 1, 15-21.
Vaishnav, P.K., Sharma, S., & Sharma, P. (2021). Analytical Review Analysis for Screening COVID-19. International Journal of Modern Research, 1, 22-29.
Response 5: Thanks for your valuable comments to improve the manuscript.
We describe the data and its sources in Section 2.2, as below:
“Monthly temperature and precipitation data were collected from five meteorological stations (Table 1) for 38 years (1980‒2018) from the Bangladesh Agricultural Research Council (BARC). The BARC team checked all the datasets for errors and inconsistencies.”
We established the reliability and accuracy of the models using statistical assessment and visual interpretations. After training and testing the models at Rajshahi station, we selected the best model for estimating SPEI. The selected model was then applied to other stations to prove its accuracy and performance. We showed the performance using statistical metrics (such as RMSE, MAE, correlation coefficient, etc. in Table 6) and visual presentations (Taylor diagram, scatter plot and box plot in Figures 4 to 6).
We have also incorporated the papers suggested from your side to improve the quality of the paper.

Reviewer 2 Report
The manuscript represents a relevant and comprehensive study considering the drought regional drought forecasts. Although the ML models are not new, still the work has merit. For modeling, the SPEIs were calculated over 4 decades using monthly rainfall and temperature data. M5P model predicted the SPEIs better than the other models with lowest RRSE, RAE, MAE and RMSE and highest R of any other model. Following revisions can help improve the work.
Although the paper has appropriate length and informative content, several parts must be improved and written in better grammar and syntax. It would be essential if authors would consider revising the organization and composition of the manuscript, in terms of the definition/justification of the objectives, description of the method, the accomplishment of the objective, and results. The paper is generally difficult to follow. Paragraphs and sentences are not well connected. Furthermore, I advise considering using standard keywords to better present the research. Use the standard keywords not more than 5. Please revise the abstract according to the journal guideline. It must be under 200 words. The research question, method, and the results must be briefly communicated. The abstract must be more informative. I suggest having four paragraphs in the introduction for; describing the concept, research gap, contribution, and the organization of the paper. The motivation has the potential to be more elaborated. You may add materials on why doing this research is essential, and what this article would add to the current knowledge, etc. The originality of the paper is not discussed well. The research question must be clearly given in the introduction, in addition to some words on the testable hypothesis. Please elaborate on the importance of this work. Please discuss if the paper suitable for broad international interest and applications or better suited for the local application? Elaborate and discuss this in the introduction.
Please elaborate on whether the reference list covers the relevant literature adequately and in an unbiased manner? Are the statistical methods valid and correctly applied? (e.g. sample size, choice of test) Are the methods sufficiently documented to allow replication studies?
State of the art needs improvement. A detailed description of the cited references is essential. Several recently published papers are not included in the review section. In fact, the acknowledgment of the past related work by others, in the reference list, is not sufficient. Consequently, the contribution of the paper is not clear. Furthermore, consider elaborating on the suitability of the paper and relevance to the journal. Kindly note that references cited must be up to date.
Elaborate on the method used and why used this method.
Limitations and validation are not discussed adequately. The research question and hypothesis must be answered and discussed clearly in the discussion and conclusions. Please communicate the future research. The lessons learned must be further elaborated in the conclusion by discussing the results to the community and the future impacts. What is your perspective on future research?
Author Response
Response to Reviewer -2 Comments
Comments and Suggestions for Authors
The manuscript represents a relevant and comprehensive study considering the drought regional drought forecasts. Although the ML models are not new, still the work has merit. For modeling, the SPEIs were calculated over 4 decades using monthly rainfall and temperature data. M5P model predicted the SPEIs better than the other models with the lowest RRSE, RAE, MAE and RMSE and the highest R of any other model. Following revisions can help improve the work.
Although the paper has appropriate length and informative content, several parts must be improved and written in better grammar and syntax. It would be essential if authors would consider revising the organization and composition of the manuscript, in terms of the definition/justification of the objectives, description of the method, the accomplishment of the objective, and results. The paper is generally difficult to follow. Paragraphs and sentences are not well connected. Furthermore, I advise considering using standard keywords to better present the research.
Response: Thank you for your comments. Your comments helped us to improve the quality of the manuscript. Details of the revisions made based on your comments are provided in point-to-point answers to your comments.
Use the standard keywords not more than 5.
Response: Thank you for your comment. Following your suggestion, we used five standard keywords in the revised manuscript.
Please revise the abstract according to the journal guideline. It must be under 200 words.
Response: We revised the abstract. The revised abstract is now less than 200 words. We also followed the journal guidelines to format the article.
The research question, method, and the results must be briefly communicated. The abstract must be more informative.
Response: We revised the abstract. We tried to mention the research need, objective, method and results in the abstract within the word limit of 200. We include more information in the abstract, including quantitative information of model performance.
I suggest having four paragraphs in the introduction for; describing the concept, research gap, contribution, and the organization of the paper. The motivation has the potential to be more elaborated. You may add materials on why doing this research is essential, and what this article would add to the current knowledge, etc. The originality of the paper is not discussed well. The research question must be clearly given in the introduction, in addition to some words on the testable hypothesis. Please elaborate on the importance of this work. Please discuss if the paper suitable for broad international interest and applications or better suited for the local application. Elaborate and discuss this in the introduction.
Response: Thank you for your comment. We rearranged the introduction according to your comment. However, we used more paragraphs to mention the concept. It is required to clearly justify the necessity of the study in northern part of Bangladesh. The last three paragraphs of the Introduction provides research gap, contribution and organization of the paper. We added the these paragraphs in the revsied manuscript according to the suggestion of the reviewer. The added paragraphs to discuss the research gap, contribution, and the organization are as below:
“Compared to the other parts of the country, the Barind tract and the Teesta floodplain regions of the northern and northwestern parts (known as North Bengal) are highly impacted by drought due to high poverty rates, dependency on agriculture, low adaptive capacity and high variability of annual and seasonal rainfall [55,56]. Drought is a recurrent event in these regions [1,57]. Over the years, drought's severity, frequency, and variability have increased in North Bengal [1,2,7,8,58–62]. Several studies indicated that droughts have significantly affected agricultural production and the natural environment in the country in recent years [60,63]. Although there have been tremendous improvements in irrigation systems in Bangladesh in recent decades, agricultural activities remain dependent on seasonal rainfall [64]. A study conducted by Islam [65] indicated that North Bengal is experiencing significant increases in rainfall variability, long seasonal-scale dry spells and numerous instances of below-normal rainfall in recent decades, significantly hampering the crop growth. Also, variability in temperature has a substantial effect on crop yields (such as rice and wheat) in North Bengal[66]. However, there is a dearth of literature regarding the evolution of drought in eastern Bangladesh.
Drought analysis and prediction need reliable rainfall and temperature data recorded for a longer period. However, an uninterrupted climate record for a longer period is not available in most of the meteorological stations of Bangladesh. This limits the drought prediction using conventional statistical and numerical models. ML algorithms could be one of the solutions for bridging the climate data gap. Furthermore, to the best of our knowledge, no research has been published in the literature that predicts the SPEI in the study region using ML approaches applied. Hence, this work aims to assess the performance of standalone models and novel hybrid ML algorithms in SPEI estimation. Additive regression, random subspace, and M5P tree models and their hybridized versions were used to predict SPEI. Agricultural drought is measured in Bangladesh using a six-month SPEI, while the water scarcities, river flow declines, and hydrological droughts are measured using the 9, 12, and 24-month SPEIs. As a result, forecasting models were developed to predict SPEI for those four-time scales over 38 years (1980‒2018). The models were developed for a single station in northwestern Bangladesh (Rajshahi), which is extremely prone to droughts, while tested at four locations, including Bogra, Rangpur, Mymensingh and Khulna.
The organization of the remaining parts of this article is as follows: Section 2 explains the study area and the data used in the study. Besides, the theories of the ML algorithms used and the construction of prediction models using ML are discussed in this section. Section 3 explains the results obtained in this study. Section 4 provides a discussion of the study's findings. Finally, conclusions drawn from the results and discussion are provided in Section 5.
Please elaborate on whether the reference list covers the relevant literature adequately and in an unbiased manner? Are the statistical methods valid and correctly applied? (e.g. sample size, choice of test) Are the methods sufficiently documented to allow replication studies?
State of the art needs improvement. A detailed description of the cited references is essential. Several recently published papers are not included in the review section. In fact, the acknowledgment of the past related work by others, in the reference list, is not sufficient. Consequently, the contribution of the paper is not clear. Furthermore, consider elaborating on the suitability of the paper and relevance to the journal. Kindly note that references cited must be up to date.
Response: We extensively reviewed literature related to drought forecasting methods using statistical and dynamic methods, the Machine learning model as an innovative way for drought forecasting, and previous studies on drought studies in the study area. We tried to cover all notable studies. We added a new paragraph in the Introduction section of the revised manuscript, where we revied the literature on recent studies on droughts in northern Bangladesh. Please see the newly added paragraph in the Introduction section. Besides, we added new references on ML algorithms for drought prediction. Please see the revised list of references.
38 years of monthly rainfall and temperature data were used for model development. The model was developed at Rajshahi station and validated at other four locations distributed over the drought-prone northern region of Bangladesh. The validation of the model at four stations was conducted for 38 years (1980-2018). The monthly data of 38 years indicate a sample size of (38×12=456). The sample size is more than enough for statistical evaluation of model performance using correlation coefficient, root-mean-square error and other statistical metrics used in this study. We have made it clear in the revised manuscript by adding the following sentence: “The model performance was evaluated at the test location for the whole study period (1980‒2018).”
Elaborate on the method used and why used this method.
Response: We added new texts in the method section to justify the methods used in this study. The added texts are as below:
“For the perdition multiscale SPEI, this study considered three tree-based algorithms, Random Subspace (RSS), Additive Regression (AR), and M5 Pruned (M5P), and their stacking hybrid forms, AR-RSS, AR-M5P, RSS-M5P, and AR-RSS-M5P. The basic concept of tree-based algorithms is fitting decision trees to different sub-samples of calibration datasets and then integrating the prediction of each tree to provide the final output. Fitting models to different sub-samples allows data decomposition in detail for better identification of complex input-output relations. The advantages of these non-parametric methods are accurate prediction, little data pre-processing like normalization or scaling, no assumptions on space distributions. A detailed description of the standalone model can be found in the study by Elbeltagi et al.[70]. Wolpert[71]proposed the use of a hybrid stacking algorithm. This method facilitates the use of ensemble algorithms, which combine two or more algorithms throughout the training period. The idea behind stacking hybrid generalization is to use first-level learners to train and predict training data sets. For the meta learner, a new training dataset was created by combining the predicted outcomes from first-level learners. Sikora et al.[72] provided comprehensive information on stacked hybrid generalization.”
Limitations and validation are not discussed adequately.
Response: The limitations and future recommendations are provided at the end of the conclusion section as below:
“This study considered only three tree-based algorithms for drought prediction. Bangladesh’s droughts can be predicted using other ML models to see how well they work. Optimization algorithms can be combined with machine learning models to improve predictability. ML algorithms can also be tested for their ability to predict droughts based on the results of multiple DIs. A nationally applicable prediction model can be created by combining data from various locations across the country.”
The research question and hypothesis must be answered and discussed clearly in the discussion and conclusions.
Response: The research question and hypothesis are mentioned and answered in the first paragraph of the Discussion section of the revised manuscript, as below:
“Droughts frequently affect the agriculture and livelihood of farmers in northern Bangladesh. Reliable forecasting of droughts is important to reduce droughts impacts in the region. However, drought analysis and prediction need reliable rainfall and temperature data recorded for a longer period, which is absent in most of the meteorological stations of Bangladesh. ML algorithms are used in this study to overcome the limitations of climate data. The present study revealed that the SPEI, the most widely used DI, can be accurately forecasted using ML models for a multi-month horizon (i.e., 6, 9, 12 and 24). SPEI-6, SPEI-9, SPEI12, and SPEI24-months models were optimized using the best subset regression analysis. Predictions of SPEI using M5P were the most accurate across all time scales. The temporal variability of SPEIs could be replicated in different parts of western Bangladesh. Water professionals and policymakers may find this model useful for making intelligent decisions.”
Please communicate the future research.
Response: The future work recommendations are provided at the end of the conclusion section as below:
“Bangladesh’s droughts can be predicted using other ML models to see how well they work. Optimization algorithms can be combined with machine learning models to improve predictability. ML algorithms can also be tested for their ability to predict droughts based on the results of multiple DIs. A nationally applicable prediction model can be created by combining data from various locations across the country.”
The lessons learned must be further elaborated in the conclusion by discussing the results to the community and the future impacts.
Response: The lessons learned from the present study are provided at the end of the discussion section as below:
“The study revealed that a hybrid ML model can significantly improve standalone ML models' performance. Longer period droughts are better predictable than shorter period droughts. This may be due to the higher variability of shorter period droughts. Droughts have a devastating impact on society and the economy.”
.
The implication of the results can benefit the community. This is also mentioned in the last paragraph of the discussion section, as below:
“Natural disasters such as drought have a devastating impact on society and the economy. The findings of this study indicate that drought forecasting models in drought-prone eastern regions of Bangladesh have the potential to mitigate the effects of drought. Drought mitigation and irrigation planning can benefit greatly from regional drought forecasts.”
What is your perspective on future research?
Response: The perspective of the future research is mentioned at the end of the conclusion as below:
“The study revealed that a hybrid ML model can significantly improve standalone ML models' performance. Longer period droughts are better predictable than shorter period droughts. This may be due to the higher variability of shorter period droughts. Droughts have a devastating impact on society and the economy.”
